# Computer-aided identification of degenerative neuromuscular diseases based on gait dynamics and ensemble decision tree classifiers

Luay Fraiwan[1,2]*, Omnia Hassanin[1]

**1** Department of Electrical and Computer Engineering, Abu Dhabi University, Abu Dhabi, UAE, **2** Department of Biomedical Engineering, Jordan University of Science and Technology, Irbid, Jordan

* fraiwan@just.edu.jo

**Data Availability Statement:** https://physionet.org/content/gaitndd/1.0.0/.

**Funding:** This work is supported by Abu Dhabi University Research Office (Grant No. 19300518).

## Abstract

This study proposes a reliable computer-aided framework to identify gait fluctuations associated with a wide range of degenerative neuromuscular disease (DNDs) and health conditions. Investigated DNDs included amyotrophic lateral sclerosis (ALS), Parkinson's disease (PD), and Huntington's disease (HD). We further performed a statistical and classification comparison elucidating the discriminative capability of different gait signals, including vertical ground reaction force (VGRF), stride duration, stance duration, and swing duration. Feature representation of these gait signals was based on statistical amplitude quantification using the root mean square (RMS), variance, kurtosis, and skewness metrics. We investigated various decision tree (DT) based ensemble methods such as bagging, adaptive boosting (AdaBoost), random under-sampling boosting (RUSBoost), and random subspace to tackle the challenge of multi-class classification. Experimental results showed that AdaBoost ensembling provided a 6.49%, 0.78%, 2.31%, and 2.72% prediction rate improvement for the VGRF, stride, stance, and swing signals, respectively. The proposed approach achieved the highest classification accuracy of 99.17%, sensitivity of 98.23%, and specificity of 99.43%, using the VGRF-based features and the adaptive boosting classification model. This work demonstrates the effective capability of using simple gait fluctuation analysis and machine learning approaches to detect DNDs. Computer-aided analysis of gait fluctuations provides a promising advent to enhance clinical diagnosis of DNDs.

## 1 Introduction

Human motion is controlled by the neuromuscular system, which comprises all muscles, sensory neurons, and motor neurons [1]. Degenerative neuromuscular disease (DNDs) arises from the degeneration or progressive loss of the function in efferent or afferent nerves. Efferent nerves are responsible for controlling voluntary muscles, while afferent nerves communicate sensory information back to the brain and the central nervous system [2]. Examples of

The funders had no role in study design, data collection and analysis, decision to publish, or preparation of the manuscript.

**Competing interests:** The authors have declared that no competing interests exist.

common DNDs include amyotrophic lateral sclerosis (ALS), Parkinson's disease (PD), and Huntington's disease (HD). ALS is a progressive condition attributed to the preferential degeneration of upper and lower motor neurons [3, 4]. As such, the disease impacts nerve cells controlling voluntary muscle control, leading to a debilitated state affecting breathing, motion, speech, eating, and even cognition [5]. PD is caused by neuron loss in the substantia nigra, a structure responsible for releasing the neurotransmitter dopamine and plays a vital role in learning, reward, and movement [6]. PD is often associated with motor symptoms, including muscle rigidity, posture instability, rhythmic resting tremors, bradykinesia, and gait festination, propulsion, and freezing [7]. HD is a genetic condition that also affects the basal ganglia and occurs explicitly due to the loss of spiny projection neurons [3]. HD's main characteristic symptom is hyperkinesia, a state of excessive restlessness leading to involuntary chorea movements. Other symptoms may include cognitive degeneration and psychiatric dysfunction [8, 9].

Studying human locomotion to diagnose DNDs shows great promise [10]. The study of human locomotion is traditionally performed using gait fluctuation analysis and aims to extract useful spatial and temporal information to quantify human motion [11]. The data recorded using typical gait measurement systems are of periodic nature. A single gait cycle consists of a sequence of spatial events attributed to the timely foot-floor contact activity. These events, namely stride, stance, and swing, can be marked from vertical ground reaction force (VGRF) signals. Typically investigated temporal attributes of gait cycle events include duration and rate. DNDs can pose significant locomotion abnormalities, reflecting on the associated gait patterns during normal walking. Accumulated studies have shown that these abnormalities are disease-specific, and thus, gait analysis can be an effective tool to differentiate and diagnose DNDs [10, 12]. For example, Ren et al. [13] used phase synchronization and conditional entropy as parameters to distinguish healthy subjects and subjects with three neurodegenerative diseases: PD, ALS, and HD. These two parameters were calculated for five pairs of time series rhythms: stance time, swing time, stride time, percentage of swing time, and stance time percentage. Another work was done by Jian-Jun et al. [14] where the Hurst exponent was used as an indicator of aging and neurodegenerative diseases. They found that the Hurt exponent of stride intervals decreases with neurodegenerative diseases and aging. In accordance, Huasdorff et al. [15] reported a significant correlation between stride interval, aging, and HD. Older subjects and HD patients had reduced stride intervals compared to healthy subjects.

Driven by the need for economic non-invasive clinical practices, the application of computer-aided human locomotion analysis to diagnose DNDs has recently gained significant research traction. Computer-aided diagnostic systems typically integrate artificial intelligence algorithms. If the data is obtained and processed appropriately, and the detection algorithm is well chosen and optimized, the elements of human expertise and error become less detrimental to the diagnosis process. On these grounds, an extensive class of previous studies was directed towards the binary classification of normal vs. pathological conditions [16–23]. Standard procedures extracted features included statistics values [16–20], recurrence quantification analysis parameters [17], fuzzy recurrence plots [18], topological motion analysis [21, 22], and left/right-foot autocorrelation and cross correlation [23]. Employed machine learning and deep learning methods included support vector machine (SVM) [16–18], least squares SVM (LS-SVM) [18], k-nearest neighbors (KNN) [16, 21], naive Bayes [21], random forest (RF) [22], decision trees [23], adaptive Neuro-Fuzzy Inference [20], multi-layer perceptron (MLP) [16], probabilistic neural network (PNN) [17], and convolutional neural network (CNN) [19].

Worth noting, the three types of DNDs discussed earlier share over-lapping motor symptoms. Thus, as targeted in this study, an efficient approach would be needed to classify these

conditions simultaneously. In accordance, a limited number of recent studies tackled multiclass classification of DNDs. In [24], Beyrami et al. used a wide range of statistical and entropy features alongside a non-negative least squares (NNLS) classifier. This approach was applied to raw short-length VGRF signals only. Lin C-W et al. [25] investigated recurrence plot and principal component analysis to transform time-domain VGRF signals into images. These images were inputted as features to a CNN model for classification. On the contrary, Alaska et al. [19] compared the performance of several classification models, namely artificial neural network (ANN), KNN, linear SVM, and RF. In the feature transformation process, extracted temporal and spectral features included independent reconstruction components, approximate entropy, standard deviation, minimum, maximum, and mean values, and the ratio of peak-magnitude to root-mean-square.

According to previous studies, deep learning-based models tend to exhibit a highly auspicious performance when classifying DNDs in both binary and multiclass contexts. In most cases, complicated preprocessing and feature engineering techniques were also used. Compared to traditional machine learning and pattern recognition methods, training and validating a reliable deep learning architecture requires significant computational resources. Typically, this process is iterative, involves multiple model parameters, and entails specialized graphical processing units. The lack of sufficient, high-quality, and comprehensive clinical data is also considered amongst the main limitations. To exploit the value of automated disease detection systems in resource-constrained settings where only small datasets and low-cost hardware devices are available, simplistic computational approaches to characterize and classify gait patterns are worth investigation.

To address the shortcomings of previous works, this study proposes a simple yet reliable computer-aided framework that simultaneously detects a wide range of DNDs based on gait dynamics. Our primary objective is to perform a comparative performance investigation for different combinations of spatiotemporal gait patterns and ensemble classification methods. To this end, we first proposed a new approach to derive spatiotemporal gait cycle time series from VGRF signals. This approach was applied to derive parameters such as stride duration, stance duration, and swing duration. Feature characterization of the VGRF signals and the spatiotemporal gait signals was based on the statistical descriptors of root mean square (RMS), variance, skewness, and kurtosis. These descriptors were applied to raw short-length signals to maximize data availability and support the proposed framework's computational efficiency. Finally, we compared the performance of various DT ensemble models based on the concepts of bagging, adaptive boosting (AdaBoost), random under-sampling boosting (RUSBoost), and random subspace. Fig 1 illustrates the DNDs detection framework employed in this study.

This paper is organized as follows. Section 2 provides a complete description of the proposed framework and the adopted methodology in this study. Section 3 presents some statistical observations on the features extracted for various disease conditions and gait signals. Moreover, it compares the performance of the ensemble classification models as applied to each of the investigated gait signals. In section 4, an in-depth discussion of the results obtained compared to other recent studies in the literature is provided and the methodological limitations of this work are highlighted. Finally, section 5 concludes this paper.

## 2 Materials and methods

### 2.1 Dataset description

In this study, we used the publicly available Physionet database for neurodegenerative diseases gait patterns [26, 27]. This dataset comprised of a total of 48 recordings spanning three different disease conditions: amyotrophic sclerosis (13 patients), Huntington's disease (20 patients),

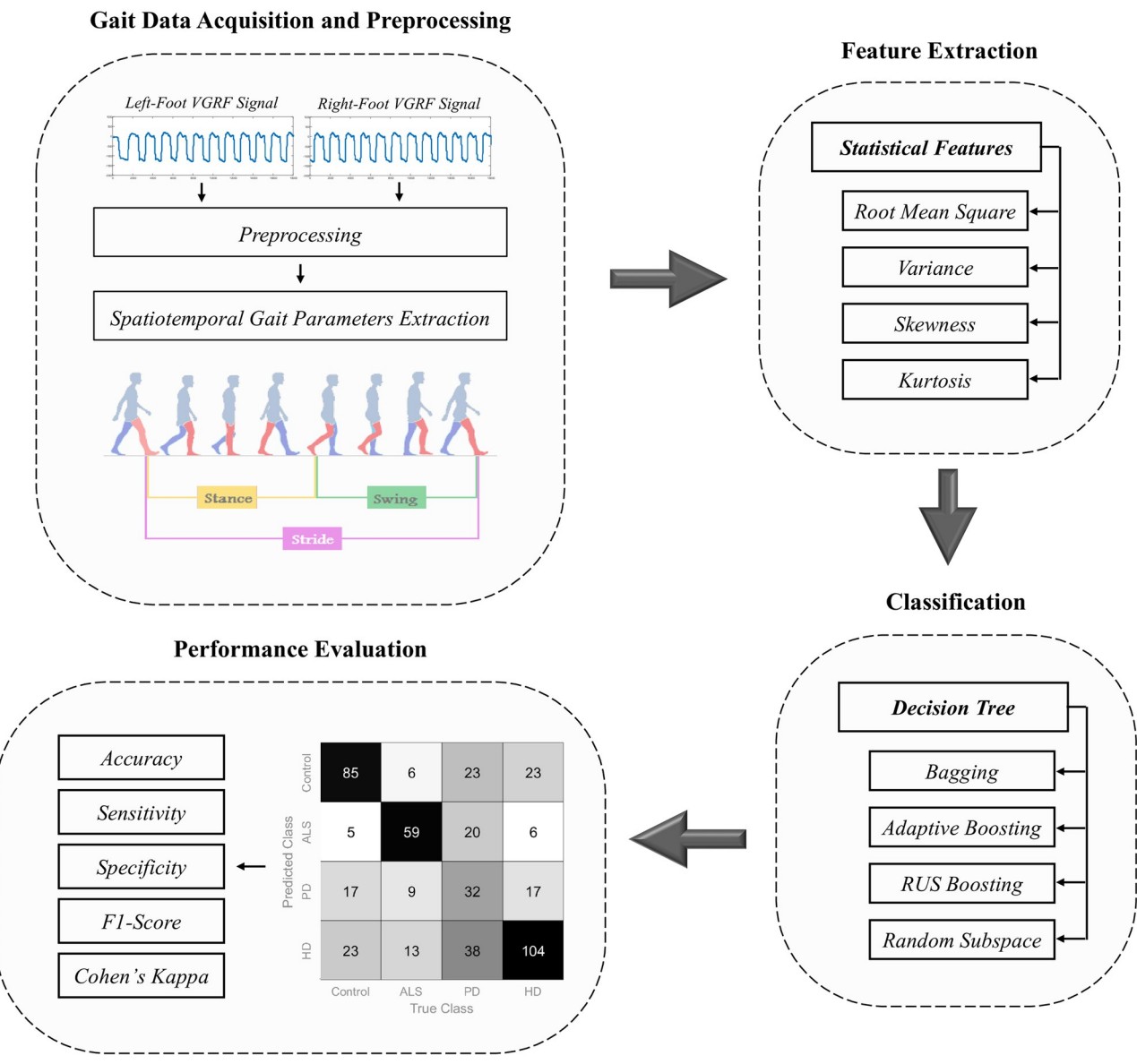

**Fig 1. Illustration of the proposed degenerative neuromuscular disease detection framework.**

and Parkinson's disease (15 patients). The dataset also included 16 healthy control subjects. Table 1 provides a characteristic and demographic summary of the subjects involved. The raw VGRF gait signals, representing the force measured under each foot separately, were recorded using eight distinct distributed force sensors under each foot (2 channels, left and right). The VGRF signals were recorded at a sampling rate of 300Hz. All the subjects were instructed to walk continuously at their average pace along a 77m hallway for 5min. When the hallway end was reached, the subjects had to turn around and walk in the opposite direction. Before data preprocessing, the data points corresponding to the first and last 15s were eliminated to reduce artifacts caused by movement start or end, as recommended by the previous work of Hausdorff et al. [26, 28]. Extreme spike values lead by the end of hallway turn-backs were corrected using a median filter [21, 29]. To maximize the number of available training instances,

**Table 1. Summary of subjects' description (average ± standard deviation values across subjects).**

| Disease Condition | No. of Subjects | Age (years) | Height (m) | Weight (Kg) | Gait Speed (m/sec) | No. of Windows |
|---|---|---|---|---|---|---|
| CON | 16 (2 M, 14 F) | 39.31 ± 18.51 | 01.83 ± 00.08 | 66.81 ± 11.08 | 01.35 ± 00.16 | 160 |
| ALS | 13 (10 M, 3 F) | 55.62 ± 12.83 | 01.74 ± 00.10 | 77.12 ± 21.15 | 01.05 ± 00.22 | 96 |
| PD | 15 (10 M, 5 F) | 66.80 ± 10.85 | 01.87 ± 00.15 | 75.07 ± 16.90 | 01.00 ± 00.20 | 125 |
| HD | 20 (6 M, 14F) | 46.65 ± 12.60 | 01.83 ± 00.11 | 72.05 ± 17.05 | 01.15 ± 00.35 | 166 |

CON: Control, ALS: Amyotrophic Lateral Sclerosis, PD: Parkinson's Disease, HD: Huntington's Disease, M: Male, F: Female.

we segmented the 5min signal recordings into multiple 30s windows without overlapping. Excessively noisy data windows were identified and discarded by manual inspection. Each window was then considered as an independent signal sample in the feature extraction and classifier training-validation process. Table 1 shows the final number of signal samples associated with each disease category after preprocessing.

**2.1.1 Extraction of spatiotemporal gait parameter signals.** In addition to the raw VGRF signals, other spatiotemporal gait parameters, such as stride duration, swing duration, and stance duration, were derived for each left and right foot independently. According to the GAITRite reference system, gait events are defined based on changes in foot-floor contact patterns. A gait cycle starts with a stance phase, during which the foot remains in contact with the ground [30, 31]. Thus, the stance duration parameter refers to the time elapsed between a heel-strike action and a subsequent toe-off action. Following is the swing phase corresponding to the stage where the foot is off-ground; the corresponding swing duration parameter bounds the toe-off action and the next gait cycle's heel-strike action. The stride combines both the stance and swing phases and corresponds to a complete gait cycle. The stride duration parameter estimates the time length of a single gait cycle marked by two successive heel-strike actions. [32, 33]

Fig 2 illustrates the stride, stance, and swing phases on the VGRF signal marked by the heel-strike and heel-off events. In order to facilitate the identification of heel-strike and toe-off points, the VGRF signals were first approximated as bilevel waveforms using the histogram methods described in [34]. At first, each VGRF signal was realized as a random variable, and the underlying probability distribution was non-parametrically constructed by binning the signal to a uniform-bin-width histogram. The appropriate histogram range and number of bins were adaptively determined for each signal. Let $A$ be a VGRF signal with a maximum amplitude $A_{max}$, a minimum amplitude $A_{min}$, the histogram range $A_R$ was calculated using:

$$A_R = A_{max} - A_{min}. \tag{1}$$

The optimal bin-width was determined using Scott's normal reference rule [35]:

$$Pin\ Width = \frac{3.49\hat{\sigma}}{\sqrt[3]{n}}, \tag{2}$$

where $\hat{\sigma}$ is the standard deviation of the signal and $n$ is the total number of time samples. Accordingly, the total number of equal-sized bins was found as:

$$M = \frac{A_R}{Pin\ Width}. \tag{3}$$

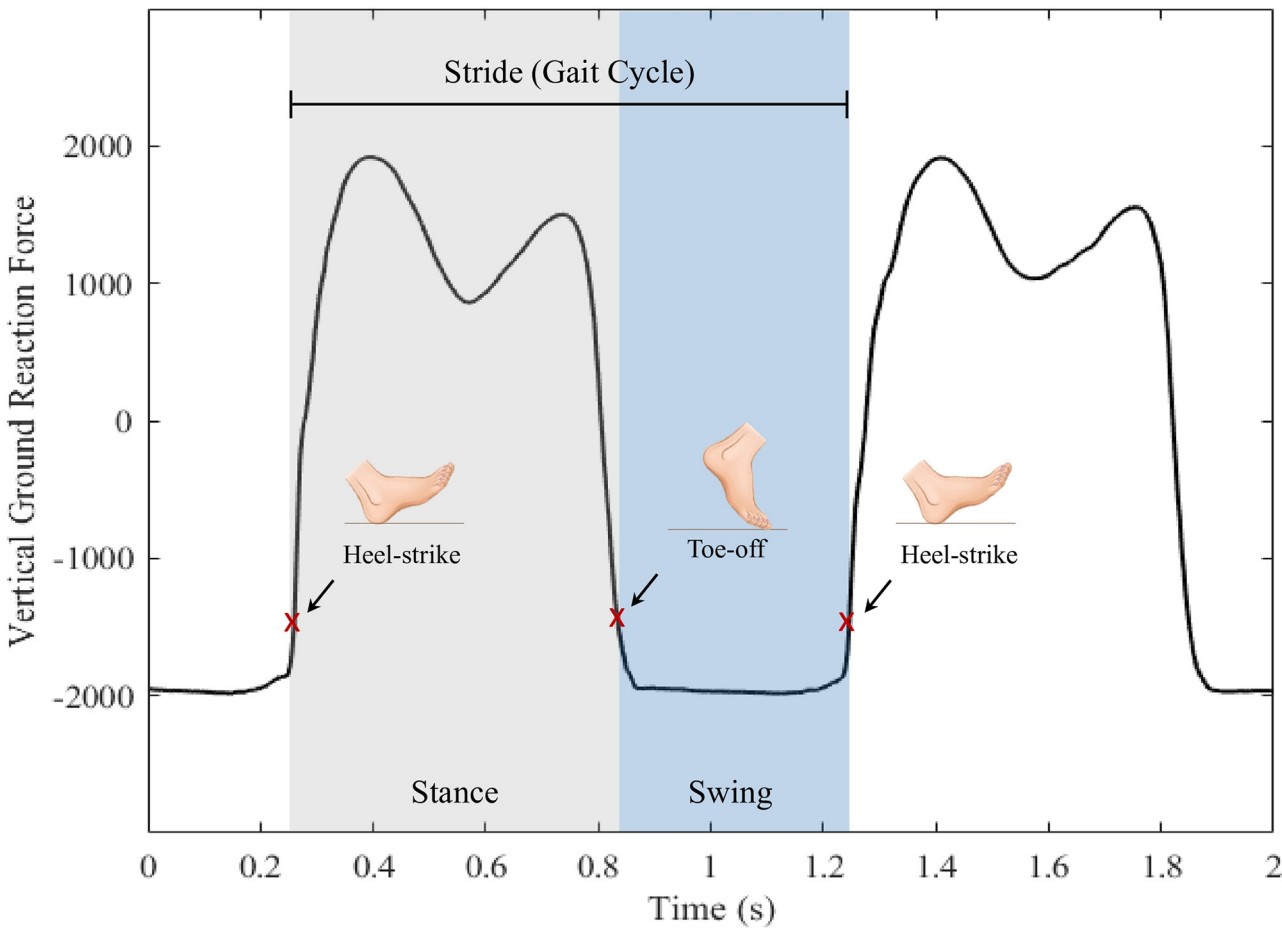

**Fig 2. Illustration the stride, stance, and swing phases on the vertical ground reaction force signal marked by the heel-strike and toe-off events.**

The constructed histogram was further divided into two sub-histograms, a lower state histogram $H_L$ with $L$ bins and an upper state histogram $H_U$ with $U$ bins, according to the following criteria:

$$L_i \mid i_{low} < i < \frac{1}{2}\left(i_{high} - i_{low}\right),$$

$$U_i \mid i_{low} + \frac{1}{2}\left(i_{high} - i_{low}\right) < i < i_{high},$$

where $i_{low}$ is the lowest index and $i_{high}$ is the highest index in the main histogram. The lower and upper state levels were then estimated as the mode of $H_L$ and $H_U$, respectively. Finally, to identify the gait events of interest, a 10% reference was set above the lower level estimated from $H_L$. For a lower bilevel $S_L$ and an upper bilevel $S_U$, the 10% reference level was set as:

$$S_L + \frac{10}{100}\left(S_U - S_L\right).$$

The heel strike point was estimated as the time instant when the positive-going transition of the VGRF signal crosses the 10% reference. Similarly, the toe-off point was estimated as the time instant when the negative-going transition crosses the same 10% reference level. Fig 3

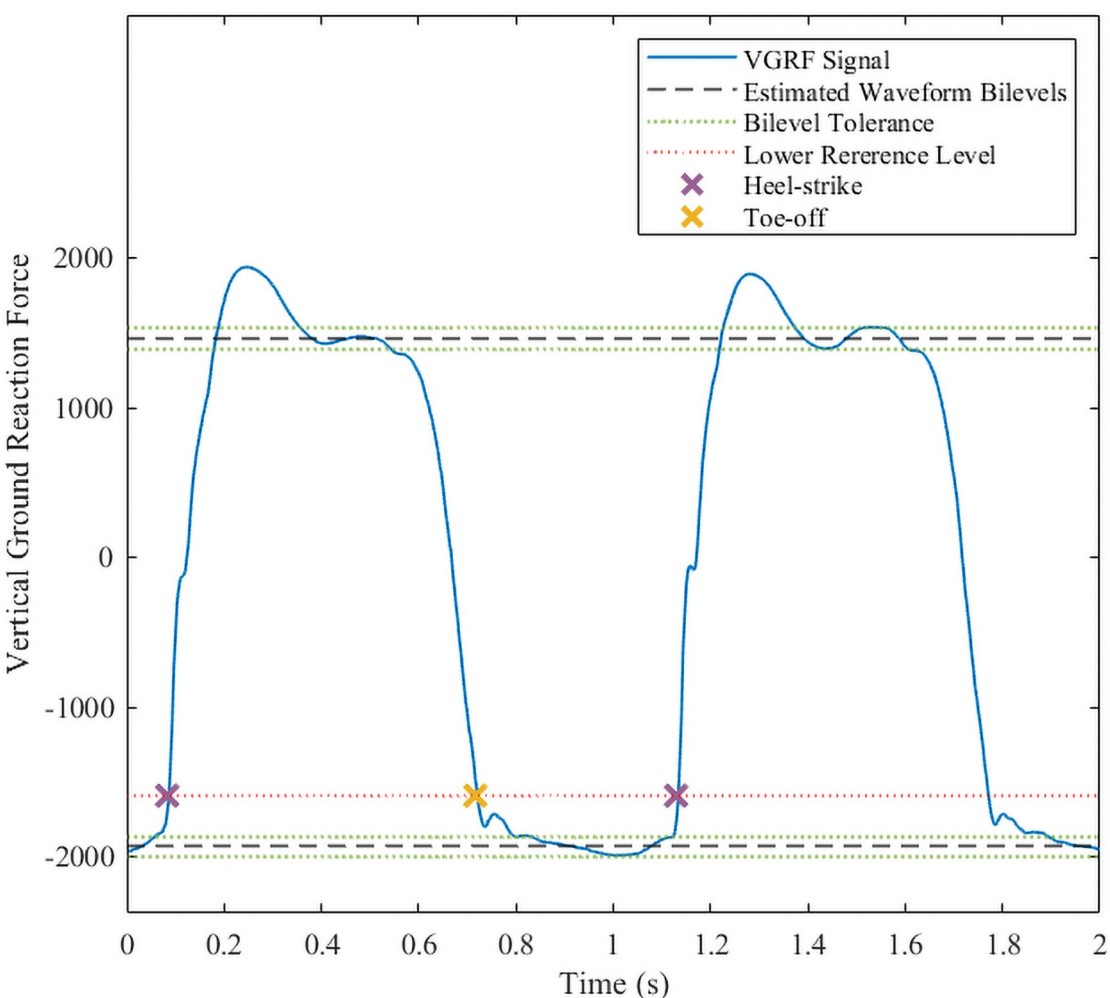

**Fig 3. Bilevel waveform estimation of the vertical ground reaction force signal to identify the heel-strike and toe-off actions time points.**

illustrates the estimated upper and lower histogram bilevels, the 10% reference level, and the corresponding heel-strike and toe-off points for a sample VGRF signal.

## 2.2 Feature extraction

The features extracted in this study included the RMS, variance, kurtosis, and skewness. These linear features provide a simple way to statistically quantify temporal changes in the amplitude, structure, and regularity of the gait signals, thus, making them an ideal option for computer-aided diagnostic tools and real-time disease detection applications.

The root-mean-square statistic (RMS) is defined as the square root of the arithmetic means of the squared of a signal $A$:

$$RMS = \frac{1}{N}\sum_{i=1}^{N}A_i^2,\qquad(4)$$

where N is the number of time samples making up the signals $A$. The variance (*var*) in statistics measures the spreadness of the signal's amplitude around its mean and is mathematically

defined as:

$$Var = \frac{1}{N-1}\sum_{i=1}^{N}(A_i - \mu_A)^2,\qquad(5)$$

where $\mu_s$ is the mean of $A$ given by:

$$\mu_A = \frac{1}{N}\sum_{i=1}^{N}A_i.\qquad(6)$$

The skewness ($Sk$) is used as a measure of amplitude asymmetry around the mean and can be computed as:

$$Sk = \frac{\frac{1}{N}\sum_{i=1}^{N}(A_i - \mu_A)^3}{Var^{\frac{3}{2}}}\qquad(7)$$

The kurtosis ($Ku$) measures the degree to which the signal distribution is prone to outliers and is calculated as:

$$Ku = \frac{\frac{1}{N}\sum_{i=1}^{N}(A_i - \mu_A)^4}{Var^2}\qquad(8)$$

The statistical temporal features were then extracted independently from each sample signal, i.e., left and right raw VGRF signals or gait parameter signals (stride, stance, and swing). The final feature vectors were formed by concatenating the statistical metrics extracted from each signal type separately. Accordingly, four distinct feature vectors, each is of size $1 \times 8$, were considered for classification.

## 2.3 Classification models

Decision Tree (DT) is a popular supervised machine learning algorithm and is amongst the most simplistic and intelligible predictive modeling approaches. As its name suggests, a DT can be thought of as a tree with root nodes, internal leaf nodes, and branches. The root nodes represent the features, the leaf nodes represent the class labels, and the branches represent the conjunctions connecting features to their class labels. The model performance depends on how well the tree is constructed from the training data. In this work, the classification and regression tree (CART) algorithm was employed to construct the DT models at the training stage [36, 37]. The Gini's diversity index was employed as the root node split criterion [38].

Different DT ensemble variations were also employed for classification, namely bagging, AdaBoost, RUSBoost, and random subspace. All investigated models were implemented following their binary realizations, and the multi-class classification problem was handled through a one-versus-all error-correcting output code ensembling. In this approach, the multi-class classification decision is made by combining the predictions of multiple base classifiers. Each base classifier performs a single binary classification task targeted towards detecting a single class from the rest [39]. The mathematical formulation of these classification methods is detailed in [40–44].

Before model training, a 10% sample subset was randomly selected from the overall dataset for tuning the classifiers' parameters. Hyperparameter tuning was done via Bayesian optimization with a cross-validation loss cost function. Table 2 summarizes the parameters selected for

**Table 2. Values of the parameters used for each classification model.**

| Classification Model | Parameter | Gait Signal | | | |
|---|---|---|---|---|---|
| | | VGRF | Stride | Stance | Swing |
| Decision Tree | Min Leaf Size | 1 | 15 | 6 | 11 |
| | Max Splits No. | 50 | 50 | 20 | 11 |
| Bagging | Learning Cycles No. | 485 | 96 | 18 | 33 |
| AdaBoost | Learning Cycles No. | 485 | 87 | 289 | 90 |
| RUSBoost | Learning Rate | 0.434 | 0.802 | 0.397 | 0.953 |
| | Learning Cycles No. | 337 | 449 | 48 | 484 |
| Random Subspace | Learning Rate | 0.500 | 0.900 | 0.700 | 0.700 |
| | Learning Cycles No. | 150 | 410 | 220 | 380 |

AdaBoost: Adaptive Boosting, RUSBoost: Random Under-sampling Boosting.

each classification model and feature vector after optimization. The complete training and validation analysis was performed via Matlab software (R2020a, Natick, Massachusetts, USA).

## 2.4 Classification performance evaluation

To get a robust estimation of the overall classification performance, the models were trained and tested using 10-folds cross-validation. To account for data imbalance, the folds were divided using an equi-stratified approach. The folds had the same number of samples (without repetition) with a class distribution following the overall dataset. The performance evaluation metrics included accuracy, sensitivity, specificity, F1-score, and Cohen's kappa coefficient ($\kappa$). Provided below are the confusion matrix-based definitions for each of these metrics:

$$Accuracy = \frac{TP + TN}{TP + TN + FP + FN} \tag{9}$$

$$Sensitivity = \frac{TP}{TP + FN} \tag{10}$$

$$Specificity = \frac{TN}{TN + FP} \tag{11}$$

$$F1 - Score = \frac{2TP}{2TP + FP + FN} \tag{12}$$

$$\kappa = \frac{Po - Pe}{1 - Pe}, \tag{13}$$

where the true positives ($TP$) and the true negatives ($TN$) represent the count of correctly classified audio signals, while the false positives ($FP$) and false negatives ($FN$) represent the number of signals incorrectly classified. $Po$ is the relative agreement between raters, and it is equivalent to the classification accuracy, while $Pe$ is the hypothetical probability of agreement by chance and can be calculated as [45]:

$$Pe = \frac{(TP + FP)(TP + FN) + (TN + FN)(TN + FP)}{(TP + TN + FP + FN)^2} \tag{14}$$

## 3 Experimental results

### 3.1 Features distribution

A one-way analysis of variance (ANOVA) was conducted to assess the significance of each statistical feature derived from each gait signal separately. The compared ANOVA levels corresponded to the disease conditions, namely control, ALS, PD, and HD. Since the feature distributions were non-normal, as revealed by the Kolmogorov-Smirnov test, a non-parametric Kruskal-Wallis ANOVA was employed. Moreover, the post-hoc Dunn-Sidák approach was used to perform pairwise comparisons between disease conditions. The tests were performed with a 95% confidence interval to verify the statistical significance of the extracted features. It is worth noting that for each feature type, uniform sample size was maintained between disease levels. A 130 sample size was selected to match the ALS category having the smallest number of samples.

Figs 4–7 visualize the features distribution for the VGRF, stride, stance, and swing signals, respectively. The $P$ and chi-square ($x^2$) values on the plots represent the results of the Kruskal-Wallis test. The asterisks represent the pairwise comparison results between disease classes after applying Dunn-Sidák correction (*:$p \leq 0.05$, **: $p \leq 0.01$, ***:$p \leq 0.001$). In general, the results positively confirmed statistical significance between different disease conditions. The investigated statistical features were highly sensitive to changes in gait dynamics between disease conditions, thus providing a promising outlook into using them for the classification analysis.

### 3.2 Classification results

Table 3 compares the performance of the investigated classification models as applied to the features derived from the VGRF, stride, stance, and swing signals independently. The tabulated values represent the average of the classification accuracy, sensitivity, specificity, F-score, and Cohen's kappa coefficient over validation folds.

**3.2.1 VGRF-based features.** Using the statistical features derived from the left and right VGRF signals, the results show that the base DT model performed poorly compared to the other ensemble classifiers with an overall average accuracy of 93.13%, sensitivity of 98.23%, specificity of 99.43%, F1-Score of 85.97% and Cohen's kappa coefficient of 81.43%. Using the AdaBoost ensemble approach, the overall performance notably improved, providing an average classification accuracy of 99.17%. Using the same classification model, the sensitivity, specificity, F1-score, and Cohen's kappa coefficient metrics reached 99.17%, 98.23%, 99.43%, 98.28%, and 97.73%, respectively. Slightly lower classification accuracies were observed for the random subspace (97.30%), Bagging (96.57%), and Boosting (96.66%) ensembles.

**3.2.2 Stride-based features.** As shown in Table 3, the AdaBoost model provided the highest detection accuracy for the stride-based feature set at an overall average of 97.68%. The DT and random subspace models a relatively lower classification accuracy of 79.06%. The highest sensitivity (58.65%), specificity (86.26%), F1-Score (58.10%) and Cohen's kappa coefficient (44.60%) were also obtained by the AdaBoost classifier. On the contrary, the Bagging ensemble model provided the worst performance, as demonstrated by its classification accuracy of 78.53%. All other metrics dropped to 56.31%, 85.34%, 55.17%, and 41.09% for the sensitivity, specificity, F1-score, and Cohen's kappa coefficients, respectively.

**3.2.3 Stance-based features.** The best classification performance for the stance-based feature set was attained using the AdaBoost classifier at an accuracy of 81.98%. Concurrently, the highest sensitivity of 63.54%, specificity of 87.81%, F1-score of 63.18% and Cohen's kappa coefficient of 51.18% were obtained using the same model. The Random substance ensemble

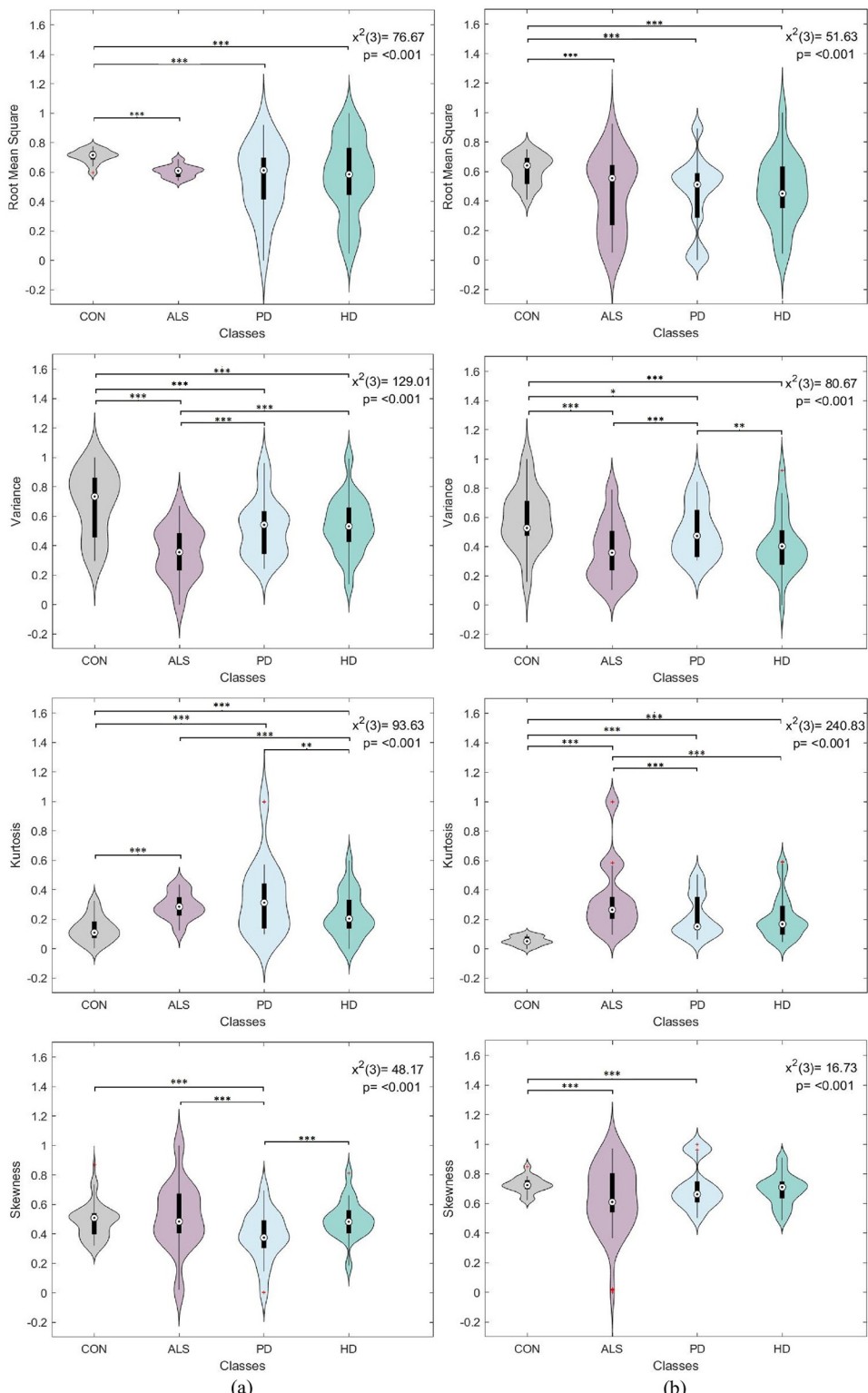

**Fig 4. Box plot and violin feature distributions for the (a) left and (b) right vertical ground reaction force signal.** The $P$ and chi-square ($x^2$) values on the plots represent the results of the Kruskal-Wallis test. The asterisks represent the pairwise comparison results between disease classes (*:$p \leq 0.05$, **:$p \leq 0.01$, ***:$p \leq 0.001$).

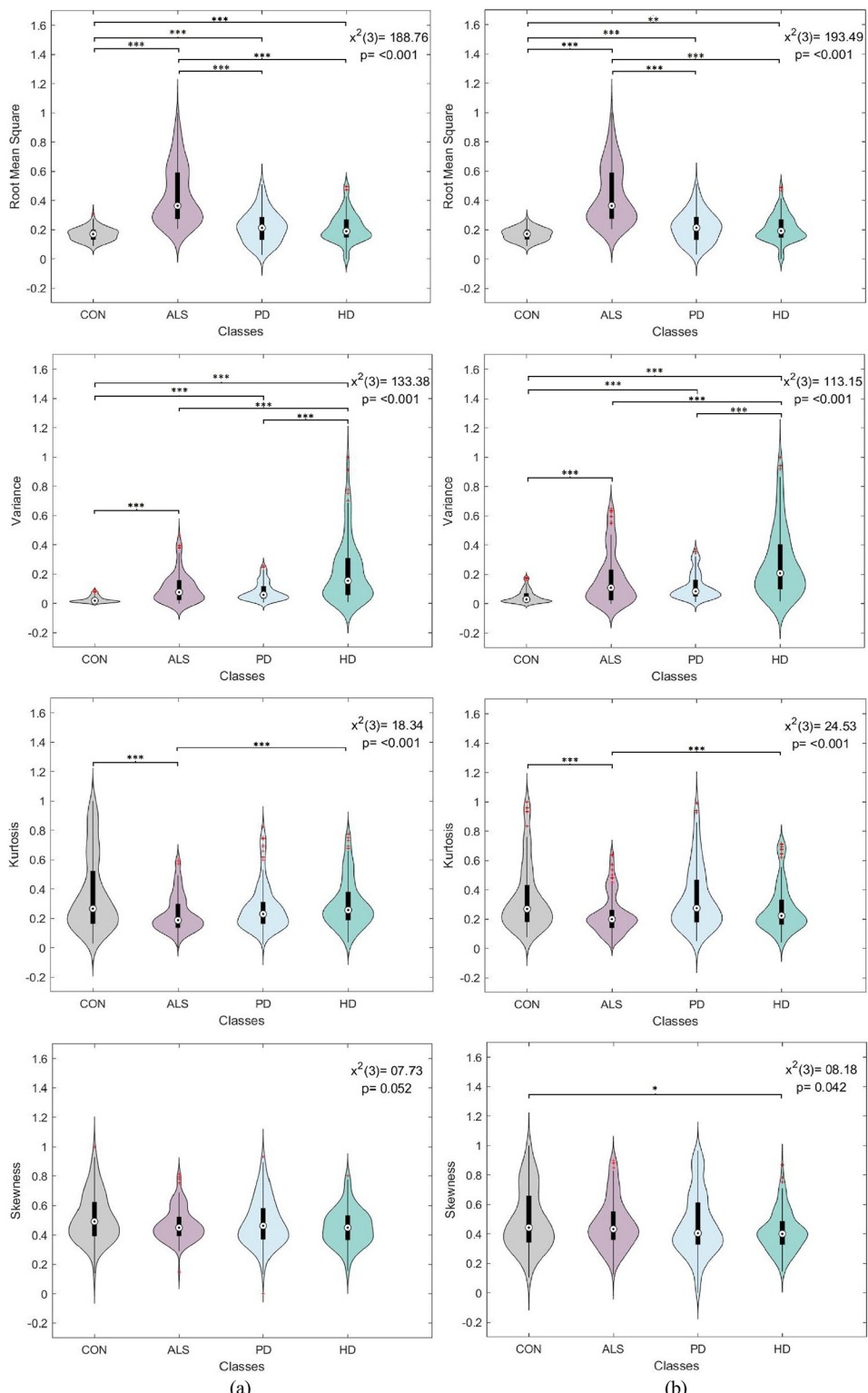

**Fig 5. Box plot and violin feature distributions for the (a) left and (b) right stride signal.** The *P* and chi-square ($x^2$) values on the plots represent the results of the Kruskal-Wallis test. The asterisks represent the pairwise comparison results between disease classes (*:$p \le 0.05$, **: $p \le 0.01$, ***:$p \le 0.001$).

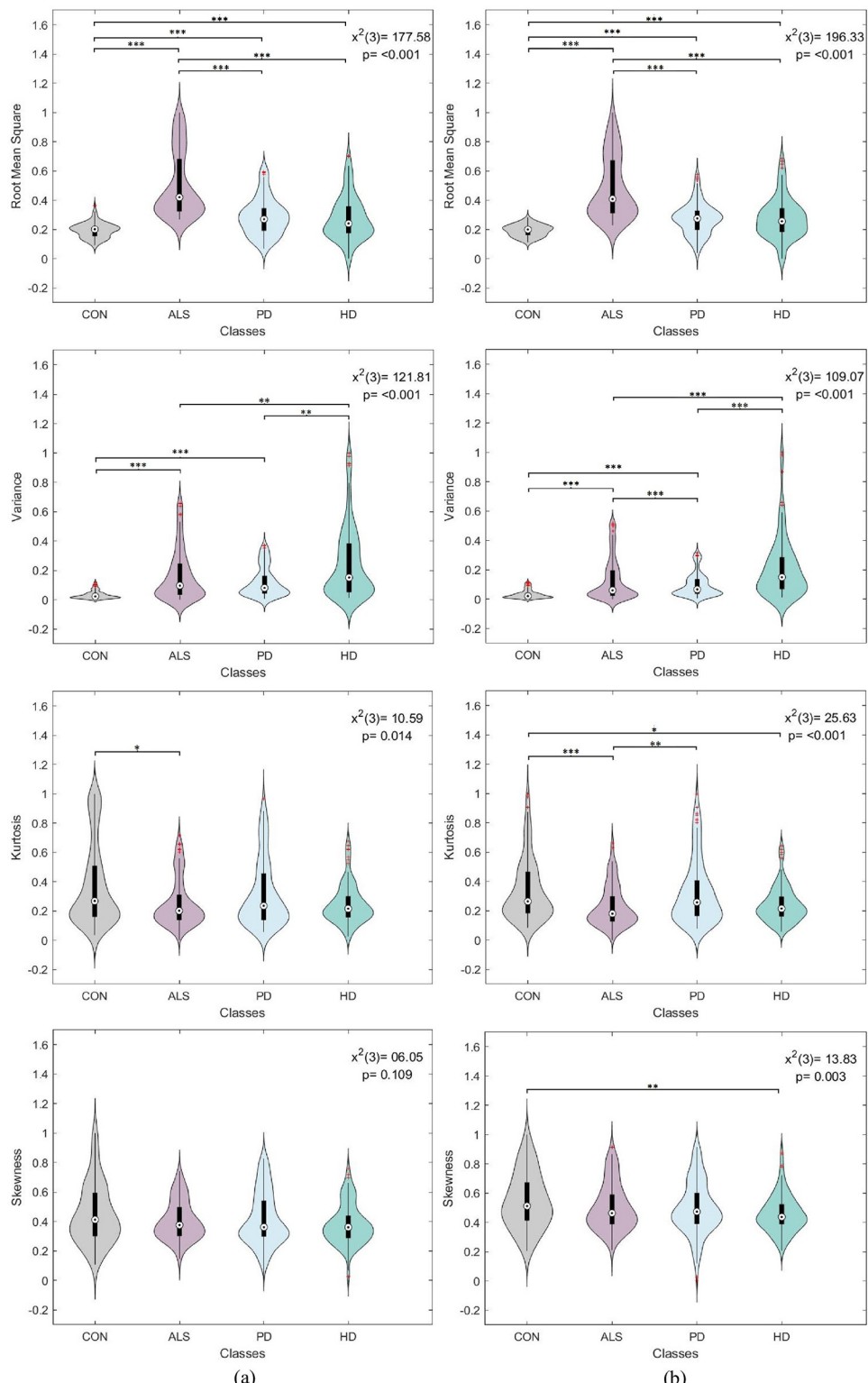

**Fig 6. Box plot and violin feature distributions for the (a) left and (b) right stance signal.** The $P$ and chi-square ($x^2$) values on the plots represent the results of the Kruskal-Wallis test. The asterisks represent the pairwise comparison results between disease classes ($^*$:$p \leq 0.05$, $^{**}$: $p \leq 0.01$, $^{***}$:$p \leq 0.001$).

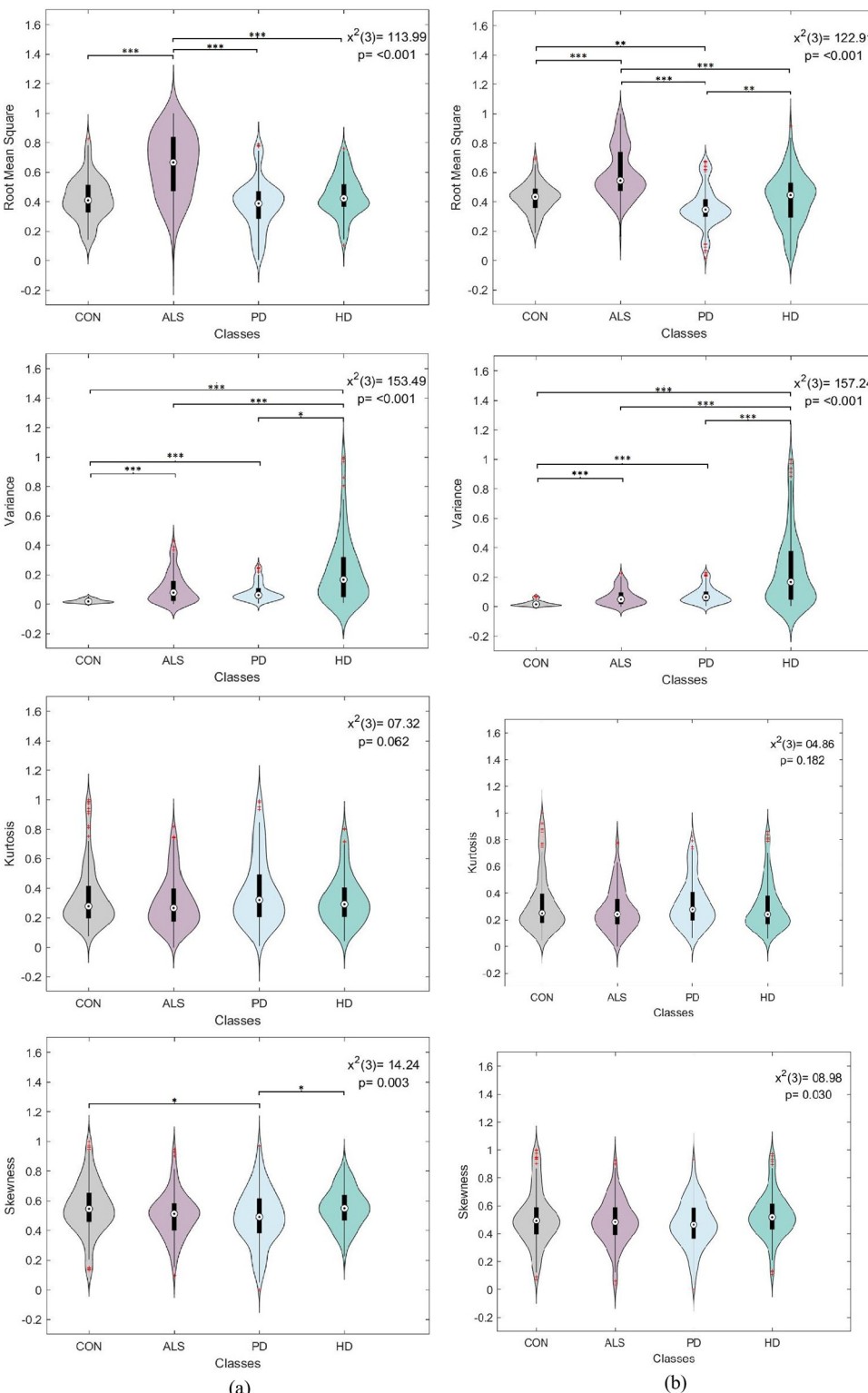

**Fig 7. Box plot and violin feature distributions for the (a) left and (b) right swing signal.** The *P* and chi-square ($x^2$) values on the plots represent the results of the Kruskal-Wallis test. The asterisks represent the pairwise comparison results between disease classes (*:$p \leq 0.05$, **: $p \leq 0.01$, ***:$p \leq 0.001$).

**Table 3. Achieved classification performance evaluation metrics for different gait signals using decision trees and different ensemble models.**

| Gait Signal | Model | Performance Criteria | | | | |
|---|---|---|---|---|---|---|
| | | Accuracy (%) | Sensitivity (%) | Specificity (%) | F1-Score (%) | Kappa $\kappa$(%) |
| VGRF | Decision Tree | 93.13 | 86.36 | 95.37 | 85.97 | 81.43 |
| | Bagging | 96.57 | 92.95 | 97.64 | 93.09 | 90.80 |
| | AdaBoost | 99.17 | 98.23 | 99.43 | 98.28 | 97.73 |
| | RUSBoost | 96.26 | 92.25 | 97.44 | 92.40 | 89.90 |
| | Random Subspace | 97.30 | 94.25 | 98.13 | 94.52 | 92.72 |
| Stride | Decision Tree | 79.06 | 57.42 | 85.74 | 55.98 | 42.35 |
| | Bagging | 78.53 | 56.31 | 85.34 | 55.17 | 41.09 |
| | AdaBoost | 79.68 | 58.65 | 86.26 | 58.10 | 44.60 |
| | RUSBoost | 78.54 | 57.52 | 85.66 | 54.77 | 41.11 |
| | Random Subspace | 79.06 | 57.42 | 85.74 | 55.98 | 42.35 |
| Stance | Decision Tree | 80.13 | 59.60 | 86.57 | 59.54 | 46.37 |
| | Bagging | 80.35 | 59.62 | 86.68 | 59.31 | 46.29 |
| | AdaBoost | 81.98 | 63.54 | 87.81 | 63.18 | 51.18 |
| | RUSBoost | 80.43 | 60.92 | 86.97 | 58.26 | 45.82 |
| | Random Subspace | 80.96 | 60.75 | 87.04 | 60.60 | 47.95 |
| Swing | Decision Tree | 77.20 | 51.38 | 84.33 | 51.02 | 36.17 |
| | Bagging | 77.20 | 49.56 | 84.08 | 47.42 | 33.38 |
| | AdaBoost | 79.30 | 55.97 | 85.71 | 56.19 | 42.51 |
| | RUSBoost | 77.73 | 53.80 | 84.89 | 53.45 | 38.78 |
| | Random Subspace | 77.71 | 52.21 | 84.63 | 51.83 | 37.28 |

VGRF: Vertical Ground Reaction Force, AdaBoost: Adaptive Boosting, RUSBoost: Random Under-sampling Boosting.

performed second to AdaBoost, followed by RUSBoost, then Bagging ensembles with average accuracies ranging between 80.96% − 80.35%. On the contrary, the base DT model provided the worst performance as evidenced by its accuracy (80.13%), sensitivity (59.60%), specificity (86.57%), F1-score (59.54%), and Cohen's kappa coefficient (46.37%).

**3.2.4 Swing-based features.** The swing-based features displayed a similar pattern to that obtained using the stride and stance features. The AdaBoost model provided a superior overall performance with an accuracy of 79.30%, sensitivity of 55.97%, specificity of 85.71%, F1-score of 56.19%, and Cohen's kappa coefficient of 42.51%. Following, the RUSBoost and random subspace ensembles demonstrated a slightly worse performance with overall overage accuracy of 77.73% and 77.71%, respectively. The worst performance among all classification models and feature sets was obtained using the bagging ensemble as evidenced by the obtained accuracy (77.20%), sensitivity (49.56%), specificity (84.08%), F1-score (47.42%), and Cohen's kappa coefficient (33.38%).

Considering that AdaBoost yielded the best improvement, it was used to gauge the effectiveness of detecting particular disease conditions. Fig 8 compares the class-specific results associated with the best performing AdaBoost model for each gait time-series signal. For the VGRF feature set, the highest class-specific accuracy was achieved for the HD classes at an average of 99.4%. The CON, ALS, and PD classes were associated with the classification accuracy of 98.8%. For the stride, stance, and swing feature sets, the highest accuracies, F1-scores, and Cohen's kappa coefficients were associated with the ALS class at the ranges of 83.75%–82.29%, 70.68%–67.68%, and 59.45%–55.49%, respectively.

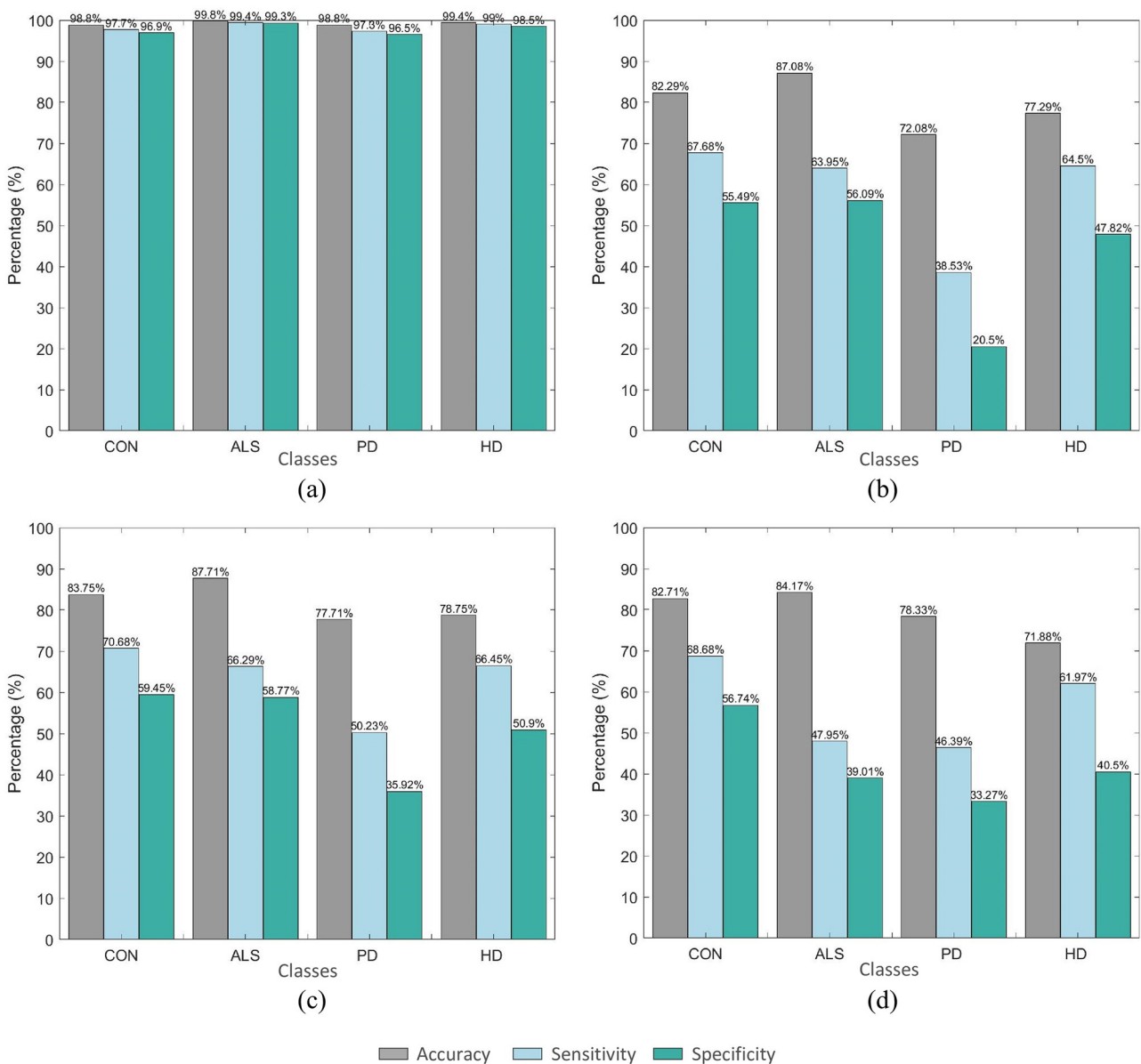

**Fig 8. Class-specific evaluation of the best performing AdaBoost ensemble model for the (a) VGRF signal, (b) stride signal, (c) stance signal, and (d) swing signal.**

## 4 Discussion

This work aimed to provide an efficient computer-assisted approach for identifying gait dynamics associated with healthy versus various DND conditions. To this end, we propose a simple yet effective framework incorporating two main stages: (1) extracting statistical temporal features from different types of gait signals and (2) and performing multi-class classification using supervised machine learning approaches. We investigated the efficiency of using ensemble learning systems, namely bagging, AdaBoost, RUSBoost, and random subspace. Moreover, we carried out a detailed statistical and classification comparison between the features extracted from different gait signals, namely left and right ground reaction force, stride, stance, and swing signals.

The prospect of machine learning usually requires extensive data transformations to provide the best possible training set to the learner. Amongst the main aspects related to data transformation is feature extraction. Optimal feature extraction provides a better representation of patterns under investigation and improves the models' predictive performance. In our proposed framework, gait signals were characterized based on statistical features, including the RMS, variance, kurtosis, and skewness. One of the main strengths of the proposed framework is using a limited number of simple features to characterize gait dynamics. These features were derived directly from raw and short-length gait singles without applying extensive preprocessing or complex filtering or transformation techniques. Such characteristic adds to the computational efficiency of the proposed framework and facilitates its application in real-time settings. Despite the simplicity, our statistical analysis showed that these features positively represented characteristic variations between disease groups. Post-hoc comparisons revealed that the features derived from the raw VGRF signals corresponded to more significant pairwise group differences.

For DND detection, we employed four types of ensemble classifiers: bagging, AdaBoost, RUSBoost, and random subspace. In order to highlight the significance of ensembled predictions, we also considered the performance of the base decision tree model. In line with the statistical analysis results, the classification analysis showed that the models' predictive performance was influenced by variability in the gait feature. Evaluation of classification performance further emphasized that the VGRF-based feature set exhibited a notably higher predictive efficiency than the other three feature sets, regardless of the classification model used. Our target of achieving a high-performance detection framework was accomplished using the AdaBoost classifier in conjugation with the VGRF-based feature set, with an average classification accuracy of 99.17%. Correspondingly, the class-specific accuracies of 98.8%, 98.8%, 98.8%, and 99.4% were achieved for the control, ALS, PD, and HD groups, respectively. Similarly, using the features extracted from the gait parameter signals, the AdaBoost model generally provided superior performance. However, we obtained a lower overall accuracy of 81.98% for the stance-based feature set, 79.68% for the stride-based feature set, and 79.30% for the swing-based feature set.

Worth noting, the classification results provided empirical evidence suggesting that ensemble classifier systems are better performers than their constituent base models. Using the base decision tree model, the VGRF-based features set provided the best classification performance, but ultimately, all ensemble techniques improved classification results to varying degrees of success. The AdaBoost yielded the most considerable improvement in all metrics, with an improvement percentage of 6.49%, 13.74%, 4.26%, 14.32%, 20.02% for the accuracy, sensitivity, specificity, F1-score, and Cohen's kappa coefficient, respectively. Following Adaboost, random subspace performed best with a 4.48% increment in the accuracy, 9.14% in the sensitivity, and 2.89% in the specificity. Slightly lower performance improvements were associated with bagging and RUSBoost models. For the gait parameter signals, the AdaBoost model showed the most notable performance improvement. The associated percentage increase in detection accuracy, sensitivity, and specificity ranged between 0.78%–2.72%, 2.14%–8.93%, and 0.61%–1.64%, respectively.

The physionet gait database was used in a few recent studies to perform a multi-class classification of neurodegenerative diseases. Table 4 provides a comparative summary of these works. In agreement with our proposed framework, adopted literature approaches generally integrated a wide range of feature extraction methods with supervised machine learning classification. The feature extraction methods for the VGRF signals included statistical amplitude quantification, detrended fluctuation analysis, and fractal dimension. The features characterizing gait parameter signals were based on statistical amplitude quantification and recurrent

**Table 4. Comparative summary to state-of-art literature on multi-class classification of neurodegenerative diseases.**

| Study | Features | Classifier(s) | Gait Signal | Performance Evaluation (Best Classifier) | | |
|---|---|---|---|---|---|---|
| | | | | Accuracy | Sensitivity | Specificity |
| Beyrami et al. (2020) [24] | Standard Deviation, Mean, Kurtosis, Approximate Entropy, Skewness | NNLS Coding | VGRF | 98.45% | – | – |
| Begum et al. (2020) [47] | **Recurrence Analysis**: Determinism, Average Diagonal Line, Recurrence Rate Entropy. **Fast Hadamard Transform**: Variance, Co-Variance, Energy, Mean, Standard Deviation | RF, SVM | Stride | 91.40% (RF) | 82.50% (RF) | 94.30% (RF) |
| Islam et al. (2019) [48] | Age, Weight, Height, Sex, Speed, Signal Maximum, Minimum, Mean, Variance, Standard Deviation, Duration Variation Coefficients, Approximate Entropy | RF | Stride, Stance, Swing | 92.39% | 90.18% | 92.61% |
| Najafabdian et al, (2018) [49] | **Independent Component Analysis**: Detrended Fluctuation Analysis, Fractal Dimension, Petrosian Fractal Dimension | AdaBoost | VGRF | 92.34% | 92.34% | 91.34% |
| Athisakthi et al. (2017) [46] | **Wavelet Transform**: Energy, Mean, Standard Deviation, Variance, Co-variance | RF, SVM | Stride | 93.75% (RF) | 91.67% (RF) | 91.67% (RF) |
| | | | | Stance | 93.74% (RF) | 87.50% (RF) | 96.15% (RF) |
| | | | | Swing | 93.75% (RF) | 87.50% (RF) | 96.15% (RF) |
| Proposed Work | Root Mean Square, Variance, Kurtosis, Skewness | DT, Bagging, AdaBoost, RUSBoost, Random Subspace | **VGRF** | 99.17% **(AdaBoost)** | 98.23% **(AdaBoost)** | 99.43% **(AdaBoost)** |
| | | | Swing | 79.68% (AdaBoost) | 58.65% (AdaBoost) | 86.26% (AdaBoost) |
| | | | Stride | 81.98% (AdaBoost) | 63.54% (AdaBoost) | 87.81% (AdaBoost) |
| | | | Stance | 79.30% (AdaBoost) | 55.97% (AdaBoost) | 85.71% (AdaBoost) |

NNLS: Non-Negative Least Squares, RF: Random Forests, SVM: Support Vector Machines, DT: Decision Tree, AdaBoost: Adaptive Boosting, RUSBoost: Random Under-sampling Boosting.

analysis. For the classification task, a limited range of standard algorithms was explored, i.e., adaptive boosting trees, random forests (RF), support vector machines, and sparse non-negative least squares (NNLS) coding. Athisakthi et al. reported the highest accuracy for the parameter gait signal through using Wavelet transform-based statistical features and RF classifier (stride 91.75%, stance 93.74%, and swing 93.7%) [46]. However, the best overall accuracy of 98.45% was obtained through statistical characterization of VGRF signals alongside NNLS coding classification [24]. Thus, it can be noted that our proposed framework significantly improved the neurodegenerative disease recognition rate in comparison to the state-of-the-art methods in the literature. Potential advantages of such an accurate diagnostic system include aiding in smart long-term monitoring. This also supports clinicians and care providers with noninvasive and low-cost tools to aid in making diagnostic decisions. A possible explanation for the relatively lower accuracies obtained using the stride, stance, and swing parameters signals might be due to the small-length raw VGRF signals used to derive these signals. However, this approach was followed since the available dataset is not large enough to perform multiple fold validation.

There may exist several methodological limitations in this study. Patient-specific factors such as age, gender, and disease severity were incongruent between different disease groups. Inconsistencies in such subject-specific factors could have a direct effect on the classification model's predictive performance. The availability of a more comprehensive dataset is essential to investigate the impact of these factors and, therefore, support the generalization ability of the proposed framework.

## 5 Conclusion

This paper investigates the application of ensemble classification to identify different DNDs. Based on normal-paced gait fluctuations, healthy and disease conditions were characterized using spatiotemporal statistical features derived from VGRF signals. For the classification task, several ensemble classification approaches were investigated based on a base decision tree classifier. A data-driven hyperparameter tuning approach using Bayesian optimization was employed to select the most proper parameter for all classification methods. The obtained results demonstrated the promising capability of detecting common DNDs, with the highest overall classification rate of 99.17%. Thus, the proposed framework is applicable to aid in diagnostic decisions while considering computing hardware resource-restricted environments. This framework can be extended in future work to include other types of DNDs and spatiotemporal gait patterns. However, this requires further experimentation spanning a broader range of subjects and disease conditions. Moreover, the investigation of other feature extraction approaches and deep learning classification models is expected to improve classification performance.

## Author Contributions

**Conceptualization:** Luay Fraiwan.

**Data curation:** Omnia Hassanin.

**Formal analysis:** Omnia Hassanin.

**Funding acquisition:** Luay Fraiwan.

**Investigation:** Luay Fraiwan.

**Methodology:** Luay Fraiwan.

**Project administration:** Luay Fraiwan.

**Resources:** Luay Fraiwan.

**Software:** Omnia Hassanin.

**Validation:** Omnia Hassanin.

**Visualization:** Omnia Hassanin.

**Writing – original draft:** Omnia Hassanin.

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
