## [Decision Letter · Decision Letter 0]

12 Apr 2021

PONE-D-21-08939

Computer-aided identification of degenerative neuromuscular diseases based on gait dynamics and ensemble decision tree classifiers

PLOS ONE

Dear Dr. Fraiwan,

Thank you for submitting your manuscript to PLOS ONE. After careful consideration, we feel that it has merit but does not fully meet PLOS ONE’s publication criteria as it currently stands. Therefore, we invite you to submit a revised version of the manuscript that addresses the points raised during the review process.

Based on the comments from the reviewers and my own observation, I recommend major revisions for this article.

We look forward to receiving your revised manuscript.

Kind regards,

Thippa Reddy Gadekallu

Academic Editor

PLOS ONE

Journal Requirements:

"NO - Include this sentence at the end of your statement: The funders had no role in study design, data collection and analysis, decision to publish, or preparation of the manuscript."

Reviewers' comments:

Reviewer's Responses to Questions

**Comments to the Author**

1. Is the manuscript technically sound, and do the data support the conclusions?

Reviewer #1: Yes

Reviewer #2: Yes

Reviewer #3: Yes

2. Has the statistical analysis been performed appropriately and rigorously? 

Reviewer #1: Yes

Reviewer #2: Yes

Reviewer #3: Yes

3. Have the authors made all data underlying the findings in their manuscript fully available?

Reviewer #1: Yes

Reviewer #2: Yes

Reviewer #3: Yes

4. Is the manuscript presented in an intelligible fashion and written in standard English?

Reviewer #1: Yes

Reviewer #2: Yes

Reviewer #3: Yes

5. Review Comments to the Author

Reviewer #1: Abstract is well-written.

No need of making separate subsections 1.1 and 1.2

Where is the related work section? Limitations in existing work should lead to the proposed work section.

Authors can consider the following recent references:

Long-term Wind Power Forecasting using Tree-based Learning Algorithms

A Consolidated Decision Tree-based Intrusion Detection System for binary and multiclass imbalanced datasets

A Trusted Social Network using Hypothetical Mathematical Model and Decision-based Scheme

Discussion and Conclusion should not be clubbed.

It is always necessary to analyze the results rather than discussing only.

Future work should be discussed with more clearity.

Write the text in simple present tense.

English editing required.

Reviewer #2: 1. Please improve overall readability of the paper.

2. I can see some paragraph in introduction, related work and proposed approach which should be merged. Same thing is with background section. Please reduce text and improve the representation of this section.

3. The objectives of this paper need to be polished. Contribution list should be polished at the end of the introduction section and last paragraph of the introduction should be the organization of the paper.

4. Introduction is poorly written. Please add background of gait fluctuations studies in the introduction.

6. Contributions at the end of introduction section should be polished.

7. Relevant literature review of latest similar research studies on the topic at hand must be discussed.

8. Quality of figures are not good.

9. Result section need to be polished.

10. All abbreviations must be described.

11. There are some grammar and typo errors.

12. Authors must add more up to date references related to security of 2020,2021 in introduction and literature.

- Authors should add these recenet references:

1) BCD-WERT: a novel approach for breast cancer detection using whale optimization based efficient features and extremely randomized tree algorithm, peerj computer science

2) Automated cognitive health assessment in smart homes using machine learning, Sustainable Cities and Society.

Reviewer #3: 1. Summarize the related works in the form of a table.

2. Some of the recent and relevant works such as the following can be discussed in the paper: "An ensemble machine learning approach through effective feature extraction to classify fake news, Early detection of diabetic retinopathy using PCA-firefly based deep learning model, An adaptive multi-layer botnet detection technique using machine learning classifiers".

3. Present a detailed analysis on theresults obtained.

4. Discuss about the limitations and future scope of the current wor.

6. PLOS authors have the option to publish the peer review history of their article (what does this mean?). If published, this will include your full peer review and any attached files.

Reviewer #1: **Yes: **Rutvij H Jhaveri

Reviewer #2: No

Reviewer #3: No

---

## [Author Response · Author response to Decision Letter 0]

14 May 2021

Al responses are included in the attached file (response to reviewers)

---

## [Decision Letter · Decision Letter 1]

17 May 2021

Computer-aided identification of degenerative neuromuscular diseases based on gait dynamics and ensemble decision tree classifiers

PONE-D-21-08939R1

Dear Dr. Fraiwan,

We’re pleased to inform you that your manuscript has been judged scientifically suitable for publication and will be formally accepted for publication once it meets all outstanding technical requirements.

Kind regards,

Thippa Reddy Gadekallu

Academic Editor

PLOS ONE

Additional Editor Comments (optional):

Reviewers' comments:

Reviewer's Responses to Questions

**Comments to the Author**

1. If the authors have adequately addressed your comments raised in a previous round of review and you feel that this manuscript is now acceptable for publication, you may indicate that here to bypass the “Comments to the Author” section, enter your conflict of interest statement in the “Confidential to Editor” section, and submit your "Accept" recommendation.

Reviewer #2: All comments have been addressed

Reviewer #3: All comments have been addressed

2. Is the manuscript technically sound, and do the data support the conclusions?

Reviewer #2: Yes

Reviewer #3: Yes

3. Has the statistical analysis been performed appropriately and rigorously? 

Reviewer #2: Yes

Reviewer #3: Yes

4. Have the authors made all data underlying the findings in their manuscript fully available?

Reviewer #2: Yes

Reviewer #3: Yes

5. Is the manuscript presented in an intelligible fashion and written in standard English?

Reviewer #2: Yes

Reviewer #3: Yes

6. Review Comments to the Author

Reviewer #2: The authors have addressed my suggestions. I would like to accept this paper.The authors have addressed my suggestions. I would like to accept this paper.The authors have addressed my suggestions. I would like to accept this paper.

Reviewer #3: I accept the paper for publication in its present form as the authors addressed all my comments in the paper.

7. PLOS authors have the option to publish the peer review history of their article (what does this mean?). If published, this will include your full peer review and any attached files.

Reviewer #2: No

Reviewer #3: No

---

## [Editor Report · Acceptance letter]

25 May 2021

PONE-D-21-08939R1 

Computer-aided identification of degenerative neuromuscular diseases based on gait dynamics and ensemble decision tree classifiers 

Dear Dr. Fraiwan:

I'm pleased to inform you that your manuscript has been deemed suitable for publication in PLOS ONE. Congratulations! Your manuscript is now with our production department. 

Kind regards, 

on behalf of

Dr. Thippa Reddy Gadekallu 

Academic Editor

PLOS ONE